# The Potential of L-Arginine in Prevention and Treatment of Disturbed Carbohydrate and Lipid Metabolism—A Review

**DOI:** 10.3390/nu14050961

**Published:** 2022-02-24

**Authors:** Aleksandra Szlas, Jakub Michał Kurek, Zbigniew Krejpcio

**Affiliations:** Department of Human Nutrition and Dietetics, Poznań University of Life Sciences, Wojska Polskiego 31, 60-624 Poznan, Poland; ola-szlas@wp.pl (A.S.); jakub.kurek@up.poznan.pl (J.M.K.)

**Keywords:** L-arginine, nitric oxide, carbohydrate metabolism, lipid metabolism

## Abstract

L-arginine, an endogenous amino acid, is a safe substance that can be found in food. The compound is involved in synthesis of various products responsible for regulatory functions in the body. Particularly noteworthy is, among others, nitric oxide, a signaling molecule regulating carbohydrate and lipid metabolism. The increasing experimental and clinical data indicate that L-arginine supplementation may be helpful in managing disturbed metabolism in obesity, regulate arterial blood pressure or alleviate type 2 diabetes symptoms, but the mechanisms underlying these effects have not been sufficiently elucidated. This review aims to present the up-to-date information regarding the current uses and health-promoting potential of L-arginine, its effects on nitric oxide, carbohydrate and lipid metabolisms, based on the results of in vivo, in vitro studies, and clinical human trials. Available literature suggests that L-arginine may have beneficial effects on human health. However, some studies found that higher dietary L-arginine is associated with worsening of an existing disease or may be potential risk factor for development of some diseases. The mechanisms of regulatory effects of L-arginine on carbohydrate and lipid metabolism have not been fully understood and are currently under investigation.

## 1. Introduction

L-arginine (2-Amino-5-guanidinovaleric acid-Arg) is an endogenous amino acid, which is mainly formed in the urea cycle. It is involved in synthesis of proteins, urea, creatine, prolamines (purtescin, spermine, spermidine), proline and nitric oxide (NO). The end products of arginine metabolism are NO, glutamate and prolamins that have various regulatory functions in the body [1,2,3,4,5,6,7,8,9,10,11,12,13,14,15].

NO is a signaling molecule, which at physiological levels stimulates glucose uptake as well as glucose and fatty acid oxidation in skeletal muscle, heart, liver and adipose tissue; inhibit the synthesis of glucose, glycogen, and fat in target tissues (e.g., liver and adipose); and enhance lipolysis in adipocytes. Thus, an inhibition of NO synthesis causes hyperlipidemia and fat accretion in rats, whereas dietary arginine supplementation reduces fat mass in diabetic fatty rats [16].

The putative underlying mechanisms on NO action were extensively presented in a number of previous publications of this topic [8,11,16,17,18,19,20,21,22,23,24,25], thus will not be repeated in detail in this review.

In short, L-arginine is a safe substance that can be found in food [20]. An adult human consumes approximately 5.4 g of arginine per day [5]. Doses of 3–8 g/d appear to be safe and not to cause acute pharmacologic effects in humans [20].

One of the causes for carbohydrate and lipid metabolism disorders in the organism may be deficiency or excess of NO. Therefore, the use of L-arginine in the prevention and treatment of lipid and carbohydrate metabolism disorders should be investigated. 

Results of recent studies indicates that L-arginine supplementation may reduce obesity, improve arterial blood pressure, mitigate oxidation and regulate endothelial dysfunction that can be helpful in managing various metabolic disorders, including type 2 diabetes mellitus (T2DM). The mechanisms of L-arginine regulatory effects may result from its contribution in stimulating lipolysis, promoting the endocrine system functions, ameliorating insulin sensitivity, restoring glucose homeostasis and fetal programming. Some in vivo studies reported that L-arginine may restore insulin sensitivity and thus mitigate T2DM [26].

This review aims to gather up-to-date information on L-arginine, an endogenous amino acid that can be used in the prevention and treatment of carbohydrate and lipid metabolism disorders for patients with T2DM and other conditions.

## 2. Transport and Absorption of L-Arginine

L-arginine is provided with food in the form of compounds with other amino acids-proteins. Absorption occurs mainly in the ileum and jejunum. Due to the high arginase activity in the small intestine, absorption of arginine in the portal vein occurs in about 60%. The remaining 40% is degraded. It is assumed that ultimately about 50% of dietary arginine enters the circulatory system [8]. Additionally, 5–15% of L-arginine in the blood comes from endogenous synthesis (from L-citrulline and L-ornithine) [5,8,11]. It can also be provided, to a lesser extent, by intracellular protein degradation [11]. 

Arginine reaches its maximum concentration in the blood approximately 2 h after oral administration. With an oral administration of 6 g, the biological half-life is 1.5–2 h [10].

Arginine is transported from the blood into the cells by specific protein transporters [9]. One such transporter is the CAT-1 protein, often found on the membrane surface together with endothelial nitric oxide synthase (eNOS). Together they form a complex responsible for the delivery of L-arginine to eNOS. This is a mechanism for direct targeting of extracellular L-arginine for NO synthesis, thus optimising its production in endothelial cells [12,13,14]. However, most of the L-arginine used to produce NO comes from the efficient recycling of L-citrulline (derived from L-arginine) back to L-arginine, which is dependent on the capacity of endothelial cells [27]. On the other hand, some studies indicate a higher importance of extracellular arginine in the production of NO [14]. A simplified pathway showing the conversion of L-arginine to NO is shown in Figure 1.

L-arginine in the liver is metabolised to creatine, urea, L-citrulline, L-ornithine, L-proline, L-glutamine and prolamins such as putrescine [1,2,3,4,15]. Under physiological conditions, L-arginine is almost completely filtered out by the glomeruli (more than 99%) [15].

## 3. Current Uses and Potential Properties of L-Arginine

L-arginine is currently used in case of liver dysfunctions (especially related to urea cycle abnormalities), ammonia poisoning, asthenic disorders and malnutrition, as well as a supportive amino acid for athletes [28]. 

It can also be administered intravenously to boost growth hormone (in case of deficiency), used in diagnostic detection of growth hormone deficiency [29,30,31,32] or as an acidifier in extreme metabolic alkalosis [29]. 

Potentially, L-arginine may find use in various types of hypertension, ischemic heart disease, heart failure, atherosclerosis (especially of the lower limbs), hypercholesterolemia, glaucoma, Raynaud’s phenomenon, chronic kidney failure, diabetes and prevention of cardiovascular disease, including stroke [20,33,34]. Despite its positive effect on many cardiovascular diseases, L-arginine supplementation may increase mortality after myocardial infarction. This effect potentially results in an increase in the production of NO, peroxynitrite, levels of NOS inhibitors and reactive oxygen species (due to impaired eNOS activity). For this reason, it is not recommended for people who has a myocardial infarction and should be used with caution in the elderly [35].

## 4. Effects of Nitric Oxide on Carbohydrate and Lipid Metabolism

NO deficiencies have been demonstrated in the course of many diseases. Among others in hypercholesterolemia and other lipid metabolism disorders, atherosclerosis and diabetes, but also in hypertension, asthma and neurodegenerative diseases [36,37,38,39,40,41,42,43,44,45]. 

The cause of NO deficiency in endothelial cells may be its degradation by active oxygen species or its reduced synthesis by the vascular endothelium. This may be due to L-arginine deficiency, cofactor deficiency, reduced nitric oxide synthase (NOS) expression or excess of endogenous NOS inhibitors (especially asymmetric dimethylarginine (ADMA)) [38,46,47]. The right amount of the cofactor is essential for proper protein expression. If there is a deficiency of a cofactor, expression may not be in sufficient quantity or quality (primarily, inadequate transport or stability of intermediates and substrate) [48]. However, it is worth noting, that although the properties of L-arginine increase the available amount of NO, there are also limitations. Research indicates that long-term use of L-arginine is ineffective in improving eNOS activity [49,50].

## 5. The Role of L-Arginine in the Nitric Acid Synthesis

L-arginine is the only endogenous substrate for NO synthesis in the human body. NO production occurs during the conversion of L-arginine to L-citrulline. The reaction takes place caused by molecular oxygen, the enzyme NOS, and various reaction cofactors. 

NOS enzymes exist in 3 isoforms. Type 1 and 3 are referred to as constitutive NOS, and are permanent components of cells. The constitutive forms produce small amounts of NO in a pulsatile manner. Type 2 is the inducible form, when activated it synthesises large amounts of NO continuously until the substrate L-arginine is depleted [51,52,53].

## 6. Potential of L-Arginine in the Treatment of Carbohydrate Metabolism Disorders

Despite conducting numerous studies, the exact effect and mechanism of action of L-arginine on carbohydrate metabolism disorders and its complications still cannot be determined. Studies often show various, contradictory results. However, the ones that can be spotted in the research most commonly and that show the greatest therapeutic potential of L-arginine are the improvement of insulin sensitivity, reduction of inflammation and oxidative stress, an increase of NO levels and vasodilation of vessels. In long-term use, L-arginine may also improve glucose tolerance and even reduce the risk of diabetes. The results of studies on the use of L-arginine in disorders related to carbohydrate metabolism are presented in Table 1.

### 6.1. Cell Testing

One of the most important organs of carbohydrate metabolism is the pancreas. For this reason, most research relies on pancreatic cell lines. 

Adeghate et al. (2001) studied the effect of L-arginine on insulin secretion by the pan-creas. Pancreas sections of type 2 diabetic and healthy rats were incubated in solutions with L-arginine. Insulin secretion in the pancreas of diseased rats was significantly stimulated by L-arginine [54]. G protein-coupled receptor family C group 6 member A (GPRC6A) is a membrane androgen receptor with a high importance for energy metabolism, hormone production and glucose homeostasis. Pi et al. (2012) demonstrated that L-arginine regulates insulin secretion through expression of GPRC6A in isolated mouse pancreatic islets. Gprc6a^−/−^ mice from which cells were harvested had metabolic syndrome including, among others, obesity and glucose intolerance. In addition, the expression of insulin was significantly lower in islets derived from Gprc6a^−/−^ mice compared with normal mice. This response was selective for insulin, as glucagon expression was not impaired in Gprc6a^−/−^ mice. GPRC6A is expressed in βcells in the pancreas, and one of its substrates is L-arginine. L-arginine provided to mouse cells with GPRC6A stimulated cAMP accumulation ex vivo through GPRC6A dependent mechanisms. L-arginine in the presence of both low and high glucose medium stimulated insulin secretion in islets of wild-type mice (with GPRC6A). L-arginine-stimulated insulin secretion rate was significantly lower in islets isolated from Gprc6a^−/−^ mice. This suggests that GPRC6A stimulated by L-arginine can regulate insulin secretion by βcells in the pancreas [55]. In contrast, Smajilovic et al. (2013) showed that GPRC6A is expressed in islets of Langerhans, but activation of this receptor by L-arginine does not stimulate insulin secretion. The Gprc6a^−/−^ mice from which the cells were collected did not show any metabolic abnormalities, compared to wild-type mice. The isolated cells, after incubation with L-arginine, did not show any changes related to insulin secretion. When glucose and L-arginine were orally administered, no changes in insulin secretion were also observed between wild-type and Gprc6a^−/−^ mice. This study suggests that GPRC6A has no essential function in blood glucose homeostasis under normal physiological conditions [56]. The differences in the results may be explained by the use of a different method of obtaining the animal model used. More specifically, the authors of the studies described above removed other exons in the process of creating Gprc6a^−/−^ mice. Different knockout models of the same gene may result in different phenotypes and therefore lead to different body responses to L-arginine. BRIN-BD11 are the β cells of the pancreas that are primarily responsible for secreting insulin in response to glucose. Their action can be disturbed, for example, by proinflammatory cytokines, which are one of the factors in the development of diabetes. Krause et al. (2011) conducted a study on the effects of L-arginine on using this cell line, in the absence or presence of proinflammatory cytokines mixture. The study showed that L-arginine at a concentration of 1.15 mM increased β cell survival by approximately 54% (to 96.3% cell survival) when exposed to proinflammatory cytokines. In the absence of exposure to cytokines, it increased survival by about 39% (to 100% cell survival). L-arginine also increased levels of glutathione and glutamate in cells, but decreased the ratio of glutathione disulphide to glutathione and decreased the release of glutamate from cells. It also stimulated higher glucose consumption and lactate production, particularly during cytokine exposure of cells. It partially attenuated the negative effect of cytokines on insulin secretion. L-arginine also stimulated acute insulin secretion, but this action was inhibited in the presence of proinflammatory cytokines. These results demonstrate the great importance of L-arginine in inflammatory reactions occurring, e.g., concerning T2DM [57].

The liver is another pivotal organ for the metabolism of carbohydrates. One of its functions is the secretion of the insulin-like growth factor (IGF-1). IGF-1 inhibits glucose synthesis in liver cells and by increasing glucose transporters synthesis it increases peripheral glucose uptake. It also reduces insulin secretion. Tsugawa et al. (2019) in a study on HepG2 liver cells and C57BL/6J mice showed that L-arginine-induced IGF-1 secretion occurs through a minimum of two mechanisms. The first is the induction of growth hormone secretion by L-arginine, which stimulates the translation and secretion of IGF-1. The second mechanism is a decrease in IGF-1 retention in the endoplasmic reticulum (ER), which leads to increased IGF-1 secretion [58]. In an experiment conducted on mouse NIT-1 pancreatic beta-cell line and the HEK293FT cell line it was found that proinsulin is retained in the ER by UDP-glucose: glycoprotein glucosyltransferase 1 (UGGT1) when arginine availability is limited. L- and D-arginine release proinsulin from UGGT1 through competition with proinsulin and promote proinsulin transfer from the ER into the Golgi apparatus. This ability has been demonstrated in several β-cell models. Therefore, it may be used in the precisely regulated insulin secretion induced by arginine [59].

A number of studies indicate that L-arginine may beneficially affect carbohydrate metabolism in various ways. The effect is mainly due to the properties that regulate insulin secretion, but also the release of IGF-1 and the mitigation of inflammation. The inconsistency of the results of a few studies indicates that the mechanisms responsible for the action of L-arginine require more investigation and explanation.

### 6.2. Animal Testing

Recently, few in vivo experiments has been done on the effects of L-arginine. This may be due to the fact that the compound is safe for administration to humans, making it available to use in clinical trials. The results of the available studies on the most important issues related to the effects of L-arginine on carbohydrate metabolism in animals are described below.

Diabetes mellitus is a disease in which a deficiency of NO and L-arginine may be observed. This was confirmed, for example, by Kohli et al. (2004). They conducted a study that showed reduced arginine levels in the blood of rats with induced diabetes. Tetrahydrobiopterin (BH_4_) and NO levels were also lower compared to the non-diabetic groups. Dietary L-arginine supplementation stimulated endothelial NO synthesis by increasing BH_4_ (essential cofactor for NO synthase) concentration. As a result of arginine supplementation, the parameters of rats with induced diabetes were similar to healthy unsupplemented rats. However, supplementation also increased the value of these parameters in healthy rats. In the case of insulin, L-arginine increased its concentration in the blood of treated rats with diabetes compared to untreated rats and in healthy supplemented rats. L-arginine supplementation reduced blood glucose levels and reduced weight loss in treated rats with diabetes compared to untreated rats [60]. Fu et al. (2005) performing a study on Zucker rats with diabetes demonstrated that a number of parameters improves after L-arginine supplementation. They found that L-arginine supplementation has great potential to increase NO synthesis and reduce body weight in case of the co-occurrence of obesity and T2DM. In addition to reducing body fat, L-arginine also decreased glucose, homocysteine, free fatty acids, dimethylarginines and leptin concentrations. L-arginine increased the expression of genes responsible for fatty acid and glucose oxidation: NO synthase-1, heme oxygenase-3, AMP-activated protein kinase, and peroxisome proliferator-activated receptor gamma coactivator-1alpha [61]. This suggests not only the hypoglycaemic effect of L-arginine, but also a positive effect on the metabolic syndrome. A hormone with pivotal role in the regulation of indices related to diabetes is glucagon-like peptide-1 (GLP-1). It affects glucose lowering by increasing insulin secretion, inhibiting glucagon secretion, reducing appetite and reducing postprandial glucose. Clemmensen et al. (2013) found that oral L-arginine supplementation acts as a stimulant of GLP-1 secretion in vivo by increasing postprandial insulin secretion and thus improving glucose tolerance. In both lean and obese mice, oral L-arginine increased plasma GLP-1 and insulin and reduced the increase in postprandial glucose. To confirm the contribution of GLP-1 receptor to these effects, L-arginine was given to Glp1r^−/−^ knockout mice and their wild-type littermates. In this test, the effect of L-arginine on wild-type mice was the same, whereas for Glp1r^−/−^ knockout littermates it showed no effect [62]. There were also studies testing whether the earlier supply of arginine could prevent the consequences of the destruction of pancreatic cells by alloxan. El-Missiry et al. (2004) proved that oral L-arginine supplementation for 7 days, both before and after treatment with alloxan, significantly reduced the thiobarbituric acid-reactive substances (TBARS) concentrations in liver and brain to values similar to the control group. Treated rats also had higher glutathione levels and higher superoxide dismutase and catalase activities in the liver and brain. Treatment with L-arginine for 7 days before alloxan injection proved to have an impact on the control of serum glucose levels compared with treatment with L-arginine for 7 days after alloxan injection. Using L-arginine either before or after alloxan injection showed similar effects on cholesterol levels. L-arginine administered to healthy rats, did not significantly affect any of the parameters [63]. Another study, conducted by Ortiz et al. (2013), investigated the physiology of mitochondria and their production of NO in cortex cells of rats with induced diabetes and the effect of L-arginine administration on these cells. Hyperglycaemia impaired mitochondrial function and increased mitochondrial release of free radicals, whereas NO and NOS-1 production decreased significantly. One could spot an increased level of TBARS in the brain cortex, which indicated lipid peroxidation. Treatment of rats with L-arginine did not reduce glucose levels, but significantly ameliorated the oxidative stress (primarily lower TBARS level), improved mitochondrial function and prevented a decrease in NO in diabetic rats [64]. Both of these studies clearly showed that TBARS, a byproduct of lipid peroxidation, is elevated in diabetes, and L-arginine supplementation significantly lowers it. Also, both studies showed that L-arginine supplementation after injection of alloxan did not affect hyperglycemia. L-arginine may also potentially minimize the risk of diabetic complications, such as tissue damage, by affecting not only proinflammatory substances but also their receptors. A study by Pai et al. (2010) showed that L-arginine supplementation of rats with T2DM significantly reduced liver and lung receptor of advanced glycation end products expression and inflammatory protein levels, which may consequently reduce tissue damage associated with T2DM. In contrast, L-arginine did not significantly affect cholesterol and glucose levels [65].

In all studies, L-arginine showed a reduction in inflammation. In contrast, the effects on glucose and insulin levels were different. Differences in action may result from different animal models, different doses of L-arginine, time and method of their administration (in water, feed or intragastrically). The above studies also show that when supplementation with L-arginine healthy animals, does not appear to have side effects.

### 6.3. Human Research

Although there are a large number of publications on clinical experiments on the effects of L-arginine, it is still not sufficient to clearly recommend or reject L-arginine as a carbohydrate regulating supplement. 

Apart from NO deficiency in the course of diseases, there may be disturbed transport of L-arginine, despite its adequate amount in the blood. Assumpção et al. (2010) showed that subjects with both obesity and metabolic syndrome have impaired L-arginine transport, despite normal plasma concentrations. Diminished L-arginine transport negatively affected insulin resistance and caused hyperinsulinaemia, additionally decreasing high-density lipoprotein cholesterol (HDL-C) and increasing triglycerides (TG) level. On the other hand, it did not affect NO production and platelet aggregation [77]. Rajapakse et al. (2013) investigated the effect of insulin on L-arginine transport in the body. Patients with T2DM and healthy controls received an infusion of radiolabelled L-arginine, followed by insulin, at a rate that did not affect blood glucose. In the control group, L-arginine led to a progressive increase in blood flow and a significant increase in NO precursor. After insulin administration, L-arginine clearance increased only in healthy subjects. This suggests that insulin resistance may significantly contribute to the onset and development of cardiovascular disease in individuals with T2DM through abnormal insulin-mediated regulation of L-arginine transport [78]. In addition to regulating insulin secretion, an appropriate level of tissue insulin sensitivity is also important. It allows the body to react appropriately to insulin and can prevent insulin resistance from turning into diabetes and slow the progression of diabetes. The study by Piatti et al. (2001) was the first to present that an increase in NO availability, due to L-arginine administration, can result in increased insulin sensitivity, even if it does not reach a normal level. L-arginine treatment improved glucose utilisation in the clamp technique by 34%, as well as endogenous glucose production by 29%. L-arginine administration in patients with T2DM significantly improved peripheral and hepatic insulin sensitivity but did not normalise it completely [67].

Obesity is a common comorbid disorder with diabetes. It is also one of the factors that increase the risk of developing diabetes. Probably clinical trials were first initiated by Wascher et al. (1997). They conducted a study on a group of healthy subjects, a group of subjects with obesity and a group of subjects with non-insulin dependent diabetes mellitus (NIDDM). The experiment was conducted twice on each subject-with a concomitant infusion of L-arginine (0.52 mg kg^−1^ min^−1^) and with an infusion without L-arginine. L-arginine restored the impaired insulin-mediated vasodilation observed in subjects with obesity and NIDDM and had no effect on healthy subjects. The L-arginine infusion improved insulin sensitivity in all three groups. However, in the insulin suppression test, the researchers observed no change in insulin, IGF-1, free fatty acids and C-peptide [66]. Patients are advised to diet and exercise properly in order to reduce body weight and improve diabetes indicators. For this reason, it is worth investigating L-arginine supplementation in such conditions as an element of supporting treatment, not just the drug itself. Such research was carried out by Lucotti et al. (2006). The study was conducted on obese, insulin-resistant type 2 diabetic patients. The subjects were divided into two groups. The first group was treated with L-arginine (8.3 g/day) and the second with placebo for 21 days. The groups followed a hypocaloric diet and an exercise training program. Body weight, waist circumference, daily glucose profile, fructosamine, insulin levels and homeostasis model assessment index were significantly reduced in both groups. L-arginine supplementation reduced waist circumference even more compared to placebo and reduced fat mass while maintaining lean body mass. It also improved daily glucose profile and fructosamine. It also increased cyclic guanosine monophosphate (c-GMP), superoxide dismutase and adiponectin levels, while it decreased endothelin-1 levels and the leptin-to-adiponectin ratio. Long-term oral L-arginine treatment compared with a diet and exercise training program alone, lead to an additive effect on glucose metabolism and insulin sensitivity. Furthermore, it improved endothelial function, oxidative stress, and adipokine release in obese, insulin-resistant type 2 diabetic patients [68]. Antioxidants play a very important role in diabetes, obesity and atherosclerosis. By catching free radicals and reducing inflammation, they prevent the worsening of these diseases and the emergence of new ones as a result of complications. The study conducted by Bogdanski et al. (2013) involved treating obese patients with L-arginine (3 × 9 g/day for 6 months) or placebo. L-arginine supplementation resulted in a significant decrease in plasminogen activator type 1, an increase in NO, total antioxidant status (TAS) and increased insulin sensitivity [72]. Suliburska et al. (2014) conducted a study on the effects of L-arginine supplementation (3 × 9 g/day for 6 months) on mineral concentration, lipid serum levels, glucose, fat content, and insulin resistance in patients with obesity. Supplementation significantly improved insulin sensitivity and increased zinc levels, which are often which are often decreased in obese subjects. It did not affect other parameters [73]. Visceral obesity further increases the risk of insulin resistance. Bogdański et al. (2012) evaluated the effects of L-arginine supplementation (3 × 9 g/day) for 3 months, on some parameters of subjects with visceral obesity. Supplementation caused a significant decrease in Homeostatic Model Assessment–Insulin Resistance (HOMA-IR) and insulin levels. In contrast, it did not affect, among others, total cholesterol (T-C), TG, glucose and TNF-α levels [70]. Lucotti et al. (2009) conducted a study on patients submitted to an aortocoronary bypass, without diabetes. Patients were treated with L-arginine (6.4 g/day) or placebo for 6 months. L-arginine decreased ADMA, improved endothelial function and increased c-GMP, L-arginine/ADMA ratio and reactive hyperemia. L-arginine, on the other hand, increased insulin sensitivity index and adiponectin, and decreased interleukin-6 and monocyte chemoattractant protein-1 levels [69]. Both of these studies suggest that impaired transport of L-arginine negatively correlates with insulin resistance and hyperinsulinemia and may contribute to cardiovascular disease, which is a common complication of diabetes. Diabetes greatly accelerates atherosclerotic processes. It is related to the intensification of inflammatory processes, as a result of which atherosclerotic plaque grows. Jabłecka et al. (2012) conducted a study on individuals with atherosclerotic peripheral arterial disease of lower extremities at Fontaine’s stage II and coexisting T2DM. L-arginine supplementation (3 × 2 g daily) for 2 months did not significantly affect fasting glucose or glycated hemoglobin (HbA1C). In contrast, NO concentration and TAS levels increased significantly [71]. Functional foods are of wide interest today, and are often more readily available than supplements and can support the health of the greater part of the population. Monti et al. (2013) checked whether L-arginine could be suitable as a food enrichment compound. They conducted an evaluation consisted of two studies aimed at evaluating the effects of L-arginine-enriched biscuits. Both studies included subjects with impaired glucose tolerance and metabolic syndrome. In the first pilot (one-day) study, patients took 6.6 g of L-arginine in the form of L-arginine-enriched biscuits, powdered L-arginine or placebo biscuits. Plasma L-arginine, NOx and c-GMP levels increased in both the biscuits and supplement cases. Perfusion Index-blood flow (PI-BF) levels also increased, indicating the effect of L-arginine even when added to the biscuits. In the second study-a clinical trial, patients consumed biscuits with 6.6 g L-arginine/day or placebo biscuits for 2 weeks. The group taking L-arginine had increased levels of L-arginine, NOx, c-GMP, PI-BF and insulin sensitivity index. In contrast, their glucose levels, proinsulin/insulin ratio and body fat mass (with no change in lean body mass) compared to the group consuming the placebo biscuits decreased [74].

A particularly important issue is the effect and safety of long-term use of L-arginine and the effects after discontinuation of its supplementation. The longest study of L-arginine use and its delayed effects was made by Monti et al. (2012 and 2018). The trial evaluated the efficacy of long-term L-arginine therapy in preventing or delaying T2DM. Patients with impaired glucose tolerance and metabolic syndrome took L-arginine (6.4 g/day) or placebo for 18 months. The follow-up was extended for an additional 12 months. During the first 18 months of taking L-arginine, the cumulative incidence of diabetes was not reduced, but the cumulative probability to become normal glucose tolerant increased. After a further 12 months of follow-up (without L-arginine treatment) these parameters were still better than in the placebo group [75]. Several years later Monti et al. (2018) investigated whether long-term L-arginine supplementation (18 months) has long-lasting (over 9 years) effects on diabetes incidence, insulin secretion and sensitivity, oxidative stress, and endothelial function. Although there were no differences in the probability of becoming diabetic between groups during the first 18 months of the study, after 9 years the cumulative incidence of diabetes was 40.6% for those previously taking L-arginine and 75.4% for those previously taking placebo. The increased levels for L-arginine supplementation that persisted after 9 years included the improvement in insulin secretion, insulin sensitivity or oxidative stress markers [76].

All studies consistently showed that L-arginine supplementation increases insulin sensitivity. It is possible that, in the case of impaired transport of L-arginine, which coexists with insulin resistance, its increased consumption allows it to be transported to its destination through as yet unknown mechanisms, which reduces insulin resistance. The well-proven antioxidant effect of L-arginine is also possible here. However, conflicting results were obtained as to the effects on glucose and insulin levels. They could be influenced mainly by different doses of L-arginine and different types of diseases and their severity in patients.

## 7. Potential of L-Arginine in the Treatment of Lipid Metabolism Disorders

Despite conducting numerous studies, the exact effect and mechanism of action of L-arginine on lipid metabolism disorders and its complications still cannot be determined. Studies often show various, contradictory results, especially concerning the effects on lipid profile. However, recent studies, particularly human research, show significant improvement in lipid profile connected with L-arginine supplementation. It may be also vital for the improvement of endothelial function, which dysfunction may lead to atherosclerotic lesions. The results of studies on the use of L-arginine in disorders related to carbohydrate metabolism are presented in Table 2.

### 7.1. Cell Testing

The number of in vitro studies investigating the effects of L-arginine in modulating lipid metabolism published to date seems to be insufficient. For this reason, the mechanism by which L-arginine would work cannot be determined. Tests are carried out on endothelial cells because its damage initiates atherosclerosis. The formation of atherosclerosis is a complex process. The resulting inflammation of the vascular walls, and consequently the formation of atherosclerotic plaque, is related, among others, to with the adherence of monocytes (in response to endothelial damage), the penetration of LDL-C (low-density lipoprotein cholesterol) into the vascular walls and the formation of foam cells. Tsao et al. (1994a) showed that chronic administration of L-arginine normalizes NO-dependent vasodilation and significantly inhibits atherogenesis in a hypercholesterolaemic rabbit model. The research on mononuclear cells proved that tissues isolated from L-arginine treated animals, compared to untreated animals, have significantly lower adhesiveness for monocytes. In tissues from rabbits receiving an NOS inhibitor, adhesiveness for monocytes was significantly increased. L-arginine supplementation had no effect on cholesterol levels. Despite this, the effect on decreasing endothelial adhesiveness is an important finding, as too much adhesiveness promotes atherosclerosis. This effect is most likely dependent on NO [79]. Zhang et al. (2020) conducted a study on endothelial cells isolated from Sprague-Dawley rats with atherosclerosis. The rats were divided into 4 groups where for 12 weeks: Group 1 (control group)–received standard food + injection of 2 mL saline, Group 2—high-fat food + injection of 2 mL saline, Group 3—high-fat food + was treated with an injection of L-arginine (1 g/kg body weight/day), Group 4—high-fat food + was treated with an injection of simvastatin (4 mg/kg body weight/day). Both L-arginine and simvastatin were found to be a preventive agent inhibiting the miR-221 expression and increasing eNOS expression in aortic endothelial cells of rats with atherosclerosis. L-arginine could prevent apoptosis induced by oxidised low-density lipoproteins in endothelial cells, which is associated with a decrease in miR-221 expression. Thus L-arginine was proved to have a weaker effect than simvastatin but also had fewer side effects [80].

The above studies indicate that L-arginine may exert some anti-atherosclerotic effects by reducing monocyte adhesion to the endothelium and by reducing the formation of foam cells (by reducing the expression of miR-221) [95].

### 7.2. Animal Testing

Hypertension and hypercholesterolaemia are among the many factors of atherosclerosis. Hypertension damages endothelial cells, while hypercholesterolaemia (especially LDL-C) leads to the formation of atherosclerotic plaques. Cooke et al. (1992) showed that L-arginine improves endothelium-dependent relaxations. This improvement was associated with reduced lesion and thickening of the vessel inner membrane in hypercholesterolaemic rabbits. L-arginine had no effect on cholesterol levels [81]. One of the complications of atherosclerosis is thrombosis. It appears as a result of a reduction in the diameter of the vessel with atherosclerotic plaque and a local reduction in the production of anticoagulants. It is associated with impaired endothelial function. Tsao et al. (1994b) conducted a study of the effects of L-arginine supplementation on platelet aggregation and cholesterol levels in rabbits. Rabbits were given a normal diet with 1% cholesterol, 1% cholesterol with L-methionine or L-arginine for 10 weeks. Cholesterol levels were not significantly different between the group supplemented with L-arginine and L-methionine and those fed the cholesterol diet alone. However, L-arginine significantly increased c-GMP levels and showed great anti-aggregation properties. Most likely this effect was due to the metabolism of L-arginine to NO [82]. Nematbakhsh et al. (2008) showed that L-arginine administered in water to hypercholesterolaemic rabbits did not affect cholesterol levels, but restored normal endothelial function. This occurs by decreasing apoptosis of endothelial cells [83]. The above studies indicate that L-arginine may influence the atherosclerotic process by acting at its various stages related to endothelial dysfunction.

Atherosclerosis is associated with disturbed lipid profile. In clinical practice it is used for disease diagnosis. Excessive levels of LDL-C has a negative effect on the vessels, increasing the cardiovascular risk, while the appropriate level of HDL-C, reduces it. Méndez and Balderas (2001) administered L-arginine to diabetic rats for 12 days by injection and took their blood samples daily to determine glucose levels, lipid profile and apolipoproteins concentrations. Concerning glucose, TG, T-C, total lipids and apolipoproteins B-100, the increase in each group was highest around day 2–6 after alloxan injection. Also, on these days, decreases in levels of LDL-C and HDL-C occurred. Administration of L-arginine significantly prevented changes, caused by alloxan, of all parameters except HDL-C. However, HDL-C levels were still significantly higher in treated rats compared to untreated rats. At the end of treatment (day 11–12), the levels of all parameters of L-arginine treated rats, except HDL-C were similar to those of the control (healthy) group. The researchers suggest that the beneficial effect of L-arginine administration on serum glucose values and lipid levels in diabetic rats may be due to the polyamine formation [84]. Aly et al. (2014) showed, in a study on rats with induced T2DM, that oral L-arginine supplementation (supplemented only after induction of diabetes and both before and after induction of diabetes) was not able to cause a significant regulatory effect in insulin sensitivity, glucose concentration, HDL-C, LDL-C, T-C and TG levels compared to untreated diabetic rats. Furthermore, in the non-diabetic groups, parameters between L-arginine-taking and non-supplemented rats did not differ [86].

L-citrulline is also important in the production of NO, so it is worth comparing its action with L-arginine. L-arginine can be produced in the body as a result of the conversion of L-citrulline, and in the liver, L-arginine is metabolized into L-citrulline. El-Kirsh et al. (2011) conducted a study on rats fed for 8 weeks with normal diet (group 1—control), normal diet + L-arginine (group 2), normal diet + L-citrulline (group 3), high cholesterol diet (group 4), high cholesterol diet + L-arginine (group 5) and high cholesterol diet + L-citrulline (group 6). Group 2 compared to the control group had significantly lower T-C, TG, VLDL-C and T-C/HDL-C ratio. Feeding a high-cholesterol diet significantly increased serum aspartate aminotransferase (AST) and alanine aminotransferase (ALT) activities, urea concentration and lipid profile parameters and significantly decreased serum HDL-C cholesterol. Serum NO levels decreased non-significantly. Administration of L-arginine or L-citrulline reversed the increase in serum AST and ALT activities, urea concentration and lipid profile parameters. At the same time, HDL-C and NO concentrations increased. Rats fed a high-cholesterol diet and treated with oral L-arginine or L-citrulline had a higher relative percentage of 18:0, 20:0 and 22:6 and lower relative percentage of 16:0 fatty acids compared to untreated rats receiving a high-cholesterol diet. Microscopic examination of the endothelium showed less adverse structural changes in group 5 and 6, compared with group 4 [85].

The lipid profile normalization studies gave contradictory results. The effects could have depended on the different doses of supplementary compounds, duration of treatment as well as the animal model used in the experiments. For instance, an improvement in the blood lipid profile was noted in rats, but not in rabbits.

### 7.3. Human Research

There have been few studies testing the regulatory potential of supplementary L-arginine on blood lipid profile in humans. The results are inconsistent or controversial. For example, Hurson et al. (1995) conducted a study on elderly patients. Patients took 30 g arginine aspartate (17 g free arginine) or placebo dissolved in syrup, for 2 weeks. Arginine significantly increased serum IGF-1 levels and improved nitrogen balance, compared to the control group. Arginine supplementation resulted in lower T-C and LDL-C levels, with no effect on HDL-C levels. Despite the high dose, no adverse effects were observed [87]. On the other hand, Clarkson et al. (1996) did not observe the effect of L-arginine on the lipid profile, despite an even higher dose and longer intervention. They conducted a study on hypercholesterolaemic young adults. The patients were supplemented with L-arginine (7 g, 3 times daily) or placebo for 4 weeks. Taking L-arginine had no effect on the lipid profile, but significantly improved endothelium-dependent vasodilation. After sublingual administration of glyceryl trinitrate (causing endothelium-independent dilation), there were no significant changes in vessel size [88]. The differences in the outcome could result from different supplementary doses, duration of treatment, as well as the age patients.

There are several reports showing that the growth hormone levels may affect the lipid profile and cardiovascular disease development. A decrease in growth hormone secretion is observed with aging, which may result in an increased risk of these diseases. In a study conducted by Blum et al. (2000) healthy postmenopausal women were orally supplemented with L-arginine (3 g, 3 times daily for 4 weeks) or placebo. L-arginine had no effect on lipid profile, catecholamines or insulin, but increased growth hormone levels [89].

There are studies that examined the effects of supplementary L-arginine combined with statins. Statins are drugs very often used in the treatment of hypercholesterolaemia. Schulze et al. (2009) examined how the administration of L-arginine in combination with statins affects patients with dyslipidemia. The patients were first given dietary advice for 6 weeks, then divided into a group receiving L-arginine (1.5 g, twice daily) or placebo. After 6 weeks, the L-arginine supplementation group took L-arginine and simvastatin (1.5 g L-arginine twice daily + 20 mg simvastatin once daily) and the placebo group took simvastatin (20 mg, once daily). This period also lasted 6 weeks. The patients received dietary advice throughout the study. Treatment with L-arginine alone had no effect on serum lipids compared with placebo. The combination of L-arginine with simvastatin led to a significantly stronger reduction in TG levels than in the group that took only simvastatin. L-arginine attenuated the increase in AST and fibrinogen that was induced by simvastatin. According to the authors, the way in which L-arginine improves the effects of statins may be through increased NO production, which leads to increased inducible lipoprotein lipase activity [90].

Obesity and overweight are often causes of an abnormal lipid profile and, consequently, atherosclerosis. Patients are advised to exercise and lose weight as part of conventional treatment. Therefore, it is worth researching the effects of L-arginine in combination with exercise. Nascimento et al. (2014) evaluated the effects of short-term L-arginine supplementation on the lipid profile and inflammatory proteins after acute resistance exercise in overweight men. Patients took 3 × 2 g of L-arginine or placebo for 7 days. L-arginine combined with resistance exercise reduced LDL-C and nonesterified fatty acids levels. Resistance exercise may have promoted this effect of L-arginine by reducing plasminogen activator inhibitor-1 levels [91]. Dashtabi et al. (2016) conducted a study on obese patients treated with L-arginine (3 or 6 g, 3 times daily) or placebo for 8 weeks. Supplementation with both lower and higher doses of L-arginine had a positive effect on changing anthropometric parameters and blood pressure values. The lower dose (3 g, 3 times daily) resulted in a decrease in body weight, blood pressure, T-C, LDL-C level and malondialdehyde (MDA) concentration compared to the results of this group before supplementation. The higher dose (6 g, 3 times daily) produced the same effects, but additionally resulted in a decrease in fasting glucose, HbA1C, TG, LDL-C and MDA levels, compared to the results of this group before supplementation. It also significantly increased HDL-C level [94].

The lowest dose of L-arginine was studied by Pahlavani et al. (2017). They evaluated the effect of low-dose L-arginine supplementation (2 g daily for 45 days) on cardiovascular disease risk factors such as lipid profile, blood glucose levels and blood pressure. Patients supplementing with L-arginine had significantly decreased fasting glucose levels and improved lipid profile (decrease in TG, T-C, LDL-C, and increase in HDL-C levels), compared with those supplementing with placebo. L-arginine had no effect on systolic or diastolic blood pressure [93]. Tripathi et al. (2012) evaluated the effect of L-arginine on serum lipids and cholesterol levels in patients with acute myocardial infarction (AMI). Serum T-C, HDL-C, LDL-C and TG levels were determined on days 1 and 15 of L-arginine administration (oral supplementation of 3 g daily). The T-C/HDL-C and LDL-C/HDL-C ratio was also calculated. Administration of L-arginine improved the lipid profile of the subjects, both patients after AMI and healthy. No side effects were reported in this study [92]. 

Apart from the therapeutic effects of supplementary L-arginine (or lack of such effects) observed in some human studies there are also reports showing possible side effects of such intervention. For example, Schulman et al. (2006) concluded that L-arginine, when added to standard postinfarction therapies (6 months of 9 g/d L-arginine supplementation) does not improve vascular stiffness measurements or ejection fraction and may be associated with higher postinfarction mortality, therefore the compound should not be recommended following acute myocardial infarction [35]. It seems that the duration of supplementation (15 days in Tripathi et al. (2012) vs. 6 months in Schulman et al. (2006)) or the dose may have an impact here. Bahadoran et al. (2016) investigated the association of regular dietary intake of L-arginine and both the incidence of coronary heart disease (CHD) and changes of blood pressure. A cohort of adults (2284) who participated in the Tehran Lipid and Glucose Study were followed for a mean of 4.7 years. Linear regression models were also used to indicate the association of L-arginine intake with changes of serum lipids and blood pressure during the follow-up. It was found that higher intake of plant derived L-arginine may have a protective effect whereas animal-derived L-arginine may be a risk factor for development of hypertension and CHD events [96]. The same group of researchers (Bahadoran et al., 2017) investigated the association of dietary L-arginine intake and the risk of chronic kidney disease in 1780 adults (men and women). Dietary intakes of total L-arginine as well as animal- and plant-derived L-arginine were assessed using the validated semi-quantitative food frequency questionnaire, at baseline. The findings suggested an adverse effect of higher intakes of L-arginine from animal sources that could be a dietary risk factor for development of kidney disease [97]. Recently Mirmiran et al. (2021) reported that there is an association between dietary L-arginine intakes and increase risk of T2DM. That interesting conclusion came out of a prospective cohort study conducted on 2139 T2DM-free adults (man and women) recruited in the Teharan Lipid and Glucose Study. The cohort was monitored for approximately 5.8 years. Daily intakes of protein and L-arginine were estimated using a validated food frequency questionnaire with 168 food item. Hazard Ratios (HRs) and 95% confidence intervals (CIs), adjusted for sex, age, smoking, diabetes risk score, physical activity levels, and total energy intakes as well as carbohydrate, fiber, fats and lysine, were calculated for L-arginine as both absolute in-take and its ratio from total protein. An increased risk of T2DM (HR = 2.71, 95% CI = 1.20–6.09) was observed among participants with higher intakes of L-arginine (median intake of >5.4 vs. 2.69 g/d). The Authors concluded that higher dietary L-arginine levels may increase risk of T2DM and it may have an independent role in T2DM development [98].

## 8. Summary

A review of the available literature suggests that L-arginine may have beneficial effects on carbohydrate and lipid metabolism. This has been evidenced by the results of numerous studies performed on in vitro and in vivo models, as well as some clinical trials involving patients with carbohydrate and lipid metabolism disorders. However, some studies involving diseased subjects and prospective studies with healthy people found that higher dietary L-arginine is associated with worsening of an existing disease or may be potential risk factor for development of some diseases (CHD, kidney disease, T2DM). The inconsistent outcome may result, among others, from potential confounding variables present in a particular experimental setting, differences in experimental models used, doses, duration of treatment. The mechanisms of regulatory effects of L-arginine on carbohydrate and lipid metabolism have not been fully understood and are currently under investigation.

## Figures and Tables

**Figure 1 nutrients-14-00961-f001:**
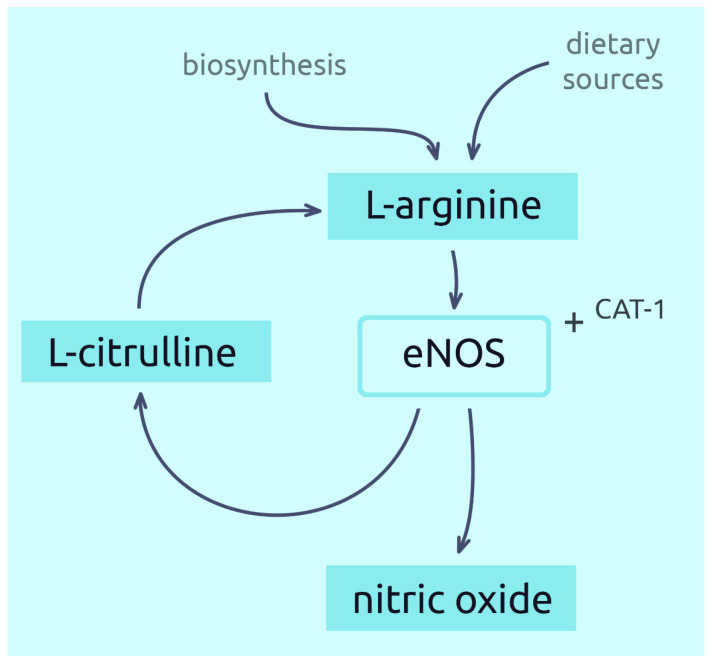
Synthesis of nitric oxide from L-arginine.

**Table 1 nutrients-14-00961-t001:** Effects of L-arginine in the treatment of carbohydrate metabolism disorders.

**Cell Testing**						
**Study (Year)**	**Cell Line/Model**	**Dose(s) of** **L-Arginine Tested**	**Control Culture**	**Outcome**		
Adeghate et al. (2001) [54]	pancreas fragments of diabetic rats	100 mM	+	L-arginine stimulates insulin secretion		
Pi et al. (2012) [55]	pancreatic islets of Gprc6a^−/−^ mice	10 mM	+	L-arginine stimulates insulin secretion in β-cells through GPRC6A activation of cAMP pathways		
Smajilovic et al. (2013) [56]	pancreatic islets of Gprc6a^−/−^ mice	20 mM	+	L-arginine induces insulin secretion, but GPRC6A is not involved in the process		
Krause et al. (2011) [57]	BRIN-BD11	0.1, 0.25, 1.15 mM	+	L-arginine induces insulin secretion, contributes to glutathione synthesis and has a protective effects in the presence of proinflammatory cytokines		
Tsugawa et al. (2019) [58]	Hep G2	1, 3.3, 10 mM	+	L-arginine increase IGF-1 level by stimulating of growth hormone secretion		
Cho et al. (2020) [59]	NIT-1 + HEK293FT	0.1, 0.2, 0.6, 1, 2 mM	+	L-arginine induces insulin secretion due to UGGT1 regulatory functions		
**Animal Testing**						
**Study, Year**	**Duration** **of Experiment**	**Dose(s) of** **L-Arginine Tested**	**Control** **Group**	**Number** **of Animals** **per Group**	**Animal** **Model**	**Outcome**
Smajilovic et al. (2013) [56]	1 min	0.05 g/kg bw intravenously + 1 g/kg bw orally	+	6–10	Gprc6a^−/−^ mice	Increase in insulin secretion after intravenous injection and oral administration of L-arginine
Tsugawa et al. (2019) [58]	120 min	3 mg/kg bw orally	+	4	C57BL/6J mice	L-arginine induces secretion of growth hormone and IGF-1
Cho et al. (2020) [59]	120 min	0.75, 1.5, 3 mg/g intraperitoneally	+	-	β cell-specific UGGT1-transgenic mice	UGGT1 mediated proinsulin management regulates insulin secretion
Kohli et al. (2004) [60]	2 weeks	0.64% in diet + 1.25% in water	+	8	Sprague-Dawley rats	L-arginine stimulates endothelial NO synthesis by increasing BH4 concentration, increased insulin concentration in the blood and reduced blood glucose level in diabetic rats
Fu et al. (2005) [61]	10 weeks	1.44% in diet + 1.25% in water	+	6	Zucker diabetic fatty rats	L-arginine increases NO synthesis, lower glucose level and reduce body weight in obese and type 2 diabetic rats
Clemmensen et al. (2013) [62]	15/120 min	1 g/kg bw orally	+	7–17	C57BL/6 mice + Glp1r^−/−^ mice	L-arginine increases GLP-1 and insulin levels and improves glucose clearance in obese mice; effects depends on GLP-1R-signaling
El-Missiry et al. (2004) [63]	1 week	100 mg/kg bw intragastrically	+	6–8	Wistar rats	L-arginine lowers serum glucose and oxidative stress in diabetic rats
Ortiz et al. (2013) [64]	4 days	622 mg/kg bw/day in water	+	5	Wistar rats	L-arginine ameliorates oxidative stress and the decrease in NO production in diabetic rats
Pai et al. (2010) [65]	8 weeks	1.5 g/kg bw/day orally	+	6–13	Wistar rats	L-arginine has no effect on plasma glucose levels, but decreases advanced glycation endproducts in diabetic rats
**Human Research**						
**Study, Year**	**Duration** **of Experiment**	**Dose(s) of** **L-Arginine Tested**	**Control** **Group**	**Number** **of Subjects** **per Group**	**Outcome**	
Wascher et al. (1997) [66]	-	0.52 mg/kg^−1^ bw/ min^−1^ (concomitant infusion)	+	7–9	L-arginine improves insulin sensitivity and restores vasodilatation (insulin-mediated) in obese and non-insulin-dependent diabetic patients; no effects was observed on insulin or IGF-1 levels	
Piatti et al. (2001) [67]	3 months (1 month of intervention)	3 × 3 g/day orally	+	12–40	L-arginine normalizes cGMP levels, improves glucose disposal and systolic blood pressure; the treatment attenuates insulin resistance in type 2 diabetic patients	
Lucotti et al. (2006) [68]	3 weeks	8.3 g/day orally	+	16–17	L-arginine positively affects glucose metabolism and insulin sensitivity, improves endothelial function, oxidative stress, and adipokine release in obese type 2 diabetic patients	
Lucotti et al. (2009) [69]	6 months	6.4 g/day orally	+	32	L-arginine regulates endothelial dysfunction, improves insulin sensitivity and reduces inflammation	
Bogdański et al. (2012) [70]	3 months	3 × 9 g/day orally	+	20	L-arginine decreases insulin level and improves insulin sensitivity; TNF-alpha plays role in the pathogenesis of insulin resistance in patients with obesity	
Jabłecka et al. (2012) [71]	2 months	3 × 2 g/day orally	+	12–38	L-arginine does not affect fasting glucose and HbA1 level in diabetic patients with atherosclerotic peripheral arterial disease, but increases NO and TAS levels	
Bogdanski et al. (2013) [72]	6 months	3 × 9 g/day orally	+	44	L-arginine decreases plasminogen activator type 1, increases NO and TAS levels, and improves insulin sensitivity in obese patients	
Suliburska et al. (2014) [73]	6 months	3 × 9 g/day orally	+	44	L-arginine affects zinc serum concentrations in obese patients; positive correlation between the change in zinc and insulin sensitivity improvement was observed	
Monti et al. (2013) [74]	6 weeks (2 weeks of intervention)	6.6 g/day orally	cross-over study	7–8/15	L-arginine improves glucose metabolism, insulin secretion and insulin sensitivity; it enhances endothelial function in patients with impaired glucose tolerance and metabolic syndrome	
Monti et al. (2012) [75]	18 months + 12-month follow-up period	6.4 g/day orally	+	72	L-arginine improves β-cell function and insulin sensitivity, and increase probability to become normal glucose tolerant, but does not reduce the incidence of diabetes in patients with impaired glucose tolerance and metabolic syndrome	
Monti et al. (2018) [76]	18 months + 90-month follow-up	6.4 g/day orally	+	45–47	L-arginine delays the development of T2DM; the effect could be related to reduction in oxidative stress	

**Table 2 nutrients-14-00961-t002:** Effect of L-arginine in the treatment of lipid metabolism disorders.

**Cell Testing**						
**Study (Year)**	**Cell Line**	**Dose(s) of** **L-Arginine Tested**	**Control Culture**	**Outcome**		
Tsao et al. (1994a) [79]	mononuclear cells of New Zealand White rabbits + WEHI 78/24	2.25% L-arginine HCl in water (animals)	+	Endothelial adhesiveness is attenuated by L-arginine; NO acts as an endogenous antiatherogenic agent; L-arginine normalizes NO-dependent vasodilation and inhibits atherogenesis in a hypercholesterolaemic rabbits		
Zhang et al. (2020) [80]	aortic endothelial cells of Sprague-Dawley rats	1 g/kg bw/day (injection to animals) + 5, 25, 50 mM (isolated cells)	+	L-arginine inhibits the expression of miR-221 and increases the expression of eNOS in cells; L-arginine exerts milder effects than simvastatin, but presumably has fewer side effects		
**Animal Testing**						
**Study, Year**	**Duration** **of Experiment**	**Dose(s) of** **L-Arginine Tested**	**Control** **Group**	**Number** **of Animals** **per Group**	**Animal** **Model**	**Outcome**
Cooke et al. (1992) [81]	10 weeks	2.25% L-arginine HCI in water	+	16–20	New Zealand White rabbits	L-arginine, as a endothelium-derived relaxing factor precursor, improves endothelium-dependent vasorelaxation
Tsao et al. (1994b) [82]	10 weeks	2.25% L-arginine HCI in water	+	3	New Zealand White rabbits	L-arginine has antiatherogenic properties and inhibits platelet aggregation in hypercholesterolaemic rabbits; the effect is presumably due to the increase in NO production
Nematbakhsh et al. (2008) [83]	4 weeks	3% L-arginine in water	+	14–16	white rabbits	L-arginine exerts no effect on T-C level, but increases nitrite concentration; L-arginine restores endothelial function in hypercholesterolaemic rabbits by the inhibition of apoptosis in endothelial cells
Méndez and Balderas (2001) [84]	12 days	10 mM/day (intraperitoneal injection)	+	5–48	Sprague-Dawley rats	L-arginine normalizes glycaemia and alleviate hyperlipidaemia by reducing TG, T-C and LDL-C levels in diabetic rats
El-Kirsh et al. (2011) [85]	8 weeks	100 mg/kg bw/day orally	+	8	albino rats	L-arginine has hypocholesterolaemic and hypolipidaemic effects; it regulates AST and ALT activities, urea level and lipid profile biomarkers; L-arginine, by promoting NO production, regulates biochemical disturbances and progression of aortic diseases; in high-fat and high-cholesterol diet fed rats.
Aly et al. (2014) [86]	8 weeks	10 mM/kg bw/day orally	+	15	Sprague-Dawley rats	L-arginine increases insulin and HDL-C levels, and decreases glucose, LDL-C, T-C and TG levels; L-arginine attenuates insulin resistance in diabetic rats
**Human Research**						
**Study, Year**	**Duration** **of Experiment**	**Dose(s) of** **L-Arginine Tested**	**Control** **Group**	**Number** **of Subjects** **per Group**	**Outcome**	
Hurson et al. (1995) [87]	2 weeks	17 g/day orally	+	15–30	L-arginine improves nitrogen balance, elevates serum IGF-1 concentrations, and reduces T-C and LDL-C levels in elderly humans; no adverse effects were observed	
Clarkson et al. (1996) [88]	12 weeks (4 weeks of intervention)	3 × 7 g/day orally	cross-over study	27	L-arginine has no effect on lipid profile (TG, T-C, HDL-C, LDL-C levels); L-arginine improves endothelium-dependent dilation in hypercholesterolaemic young adults, which might attenuate atherogenic processes	
Blum et al. (2000) [89]	3 months (1 month of intervention)	3 × 3 g/day orally	cross-over study	10	L-arginine increases growth hormone level, but does not affect insulin, catecholamines and lipid profile (TG, T-C, HDL-C, LDL-C, VLDL-C levels) in postmenopausal women	
Schulze et al. (2009) [90]	18 weeks (6 weeks of intervention)	2 × 1.5 g/day orally	+	11–22	L-arginine + simvastatin reduces TG level compared to placebo + simvastatin; L-arginine attenuates increases in AST and fibrinogen induced by simvastatin; L-arginine intensifies effects of simvastatin on lipid metabolism markers, but it has no effects when given alone in patients with hypertriglyceridaemia	
Nascimento et al. (2014) [91]	3 weeks (1 week of intervention)	3 × 2 g/day orally	cross-over study	7	No effects on TG, T-C and adiponectin levels were observed; L-arginine decreases LDL-C and non-esterified fatty acids levels; L-arginine can enhance effects of exercise inducing changes in lipid profile in overweight men	
Tripathi et al. (2012) [92]	15 days	3 g/day orally	+	60–70	L-arginine administration was found to improve the lipid profile in patients with acute myocardial infarction; L-arginine regulates modified cholesterol levels and increases HDL-C; L-arginine might be useful against precipitation of myocardial ischemia in elderly population	
Pahlavani et al. (2017) [93]	45 days	2 g/day orally	+	28	L-arginine improves glycaemia and lipid profile (TG, T-C, LDL-C, HDL-C), but has no effect on blood pressure in male athletes	
Dashtabi et al. (2016) [94]	8 week	3 × 3 or 6 g/day orally	+	27–28	L-arginine decreases, blood pressure, glycaemia, MDA, TG, T-C, LDL-C and levels and increases HDL-C level; L-arginine improves anthropometric parameters, blood pressure and blood biochemical indices in patients with obesity	
Schulman et al. (2006) [35]	6 months	3 × 3 g/day orally	+	28–30	L-arginine does not improve measurements related to vascular stiffness or ejection fraction; supplementary L-arginine might be associated with higher postinfarction mortality and should not be recommended for elderly patients after acute myocardial infarction	

## Data Availability

Not applicable.

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
