# Peer review of "The Potential of L-Arginine in Prevention and Treatment of Disturbed Carbohydrate and Lipid Metabolism—A Review"

_nutrients, 2022, doi:10.3390/nu14050961_

Round 1
Reviewer 1 Report
Szlas et al. review the application of arginine as a therapeutic to treat dysregulated carbohydrate and lipid metabolism. Overall, the review is informative and well organized. Comments and suggestions follow below.
Major comments:
- The review covers a lot of studies and may be difficult to navigate for investigators interested in specific pathways or tissues.
A figure or table summarizing molecular findings including information such as the genes or pathways found to be impacted by arginine and the tissue or cell line the study was performed on would be helpful.
A figure or table describing the clinical studies would also be helpful.
- To achieve better cohesion, the authors could organize paragraphs within sections by topic instead of having a separate paragraph for each reference. This organization appears to already be present by the order of the paragraphs. For example, the first 3 paragraphs under section 7.2
discuss endothelium, atherosclerosis, and cholesterol. These paragraphs could be combined with an introductory statement linking the studies. Similarly, the paragraphs discussing GPRC6A could be combined with an introductory statement linking the two studies.
Minor comments:
- The second and third paragraphs under section 6.1 describe two contradictory studies. Is it possible to offer a potential explanation for the contrasting findings?
- As written, line 117 states insulin is secreted by GPRC6A. Perhaps the authors intend to communicate arginine regulates insulin secretion through expression of GPRC6A.
- On line 93, the authors mention cofactor deficiency and NOS expression as potential mechanisms of NO deficiency. The authors should mention that cofactor and protein expression are related, citing a review on the subject (Jeong and Vacanti, Nutrition and Metabolism, 2020).
- Be careful that the text description of the findings from reference 26 is different enough from that written in the publication to not need quotation marks.
Potential Typos:
- Lines 59 and 60: the authors may have intended to communicate that 60% of arginine is absorbed in the portal vein
- Line 243: delete “not known”
Author Response
Thank you for the in-depth analysis of our manuscript.
All the comments have been taken into account and the content of the article has been modified to make it easier to navigate through it and achieve better cohesion.
A figure has been added to the content of the article to visualize the simplified pathway of conversion of L-arginine to nitric oxide. Also, tables summarizing described publications have been added.
The above-mentioned issues have been discussed in more detail.
The issues described in Minor comments and Potential Typos were also addressed.
Reviewer 2 Report
The authors provide a comprehensive summary of the effects of L-arginine on carbohydrate and lipid metabolism in the context of metabolic diseases such as obesity and Type 2 Diabetes. The authors did a good job summarizing a large number of studies assessing the metabolic functions of L-arginine in a wide range of systems including cells, animals and humans. It makes a lot of sense organizing the manuscript around carbohydrate and lipid metabolism the way that the authors did. However, the authors failed to integrate the results of similar studies providing implications of the work. Rather, the authors summarized study after study, one paragraph at a time without including any implications of the work making the manuscript rather disjointed and lacking flow. I think that the authors really need to rework the text describing similar studies together pointing out the similarities and differences of the results (and where there are differences, speculate why the results might be different). Integrating consistent results of similar studies will allow the biology of L-arginine to be highlighted (I believe the biological/metabolic functions of L-arginine get lost in the disjointed nature of the current manuscript). What are the major findings that have arisen from all these studies and what do they mean? I think focusing on these major findings that are consistent across multiple studies will help make this manuscript more impactful to the field. In addition, I also noticed some grammatical and punctuation errors in the text, so I would recommend the authors proofread the manuscript.
Author Response
Thank you for the in-depth analysis of our manuscript.
The content of the article has been reorganized to make it more consistent and integrated.
The paragraphs were harmonized and combined with conclusions resulting from the similarities in the described publications.
Also, grammatical and stylistic corrections were made to improve the linguistic quality of the text.
Reviewer 3 Report
The paper of Szlas et al. is an interesting review on the potential role of L-arginine in lipid and carbohydrate metabolism, with potential beneficial effects on obesity, type 2 diabetes and blood pressure.
Only a few details to correct:
Line 47: the word "diseases" is left over
Line 50: Instead of "ameliorating insulin resistance" it should indicate ""decreasing insulin resistance" or "ameliorating insulin sensitivity"
Line 102:´Should indicate "Type 2 is the inducble form, when actvated it synthesises..."
Line 117: Authors should minimally describe what GPRC6A is
Lines 383-386 and 444-448: If the observed changes are not significant, they should not be reported as a decrease. From a statistical point of view there is no variation in the parameters
Author Response
Thank you for the in-depth analysis of our manuscript.
All the comments have been taken into account and addressed.
Round 2
Reviewer 1 Report
All of our comments have been addressed.
Reviewer 2 Report
I would like to thank the authors for addressing my comments. The manuscript flows much better and provides context for the studies described as well as for any inconsistencies between studies. I also really like the addition of the tables summarizing the studies describing the roles of L-arginine in carbohydrate metabolism and lipid metabolism. I only noticed a few grammatical errors in the manuscript, but besides that, my concerns have been addressed.